# Numerical Algorithms for Computing an Arbitrary Singular Value of a Tensor Sum †

Asuka Ohashi [1,*] and Tomohiro Sogabe [2]

1 National Institute of Technology, Kagawa College, Takuma Campus, Mitoyo 769-1192, Japan
2 Department of Applied Physics, Nagoya University, Furo-cho, Chikusa-ku, Nagoya 464-8603, Japan; sogabe@na.nuap.nagoya-u.ac.jp
* Correspondence: ohashi-a@dg.kagawa-nct.ac.jp
† This paper is an extended version of our paper published in International Conference on Mathematics and Its Applications in Science and Engineering (ICMASE2020), Ankara Haci Bayram Veli University, Turkey (Online), 9–10 July 2020.

**Abstract:** We consider computing an arbitrary singular value of a tensor sum: $T := I_n \otimes I_m \otimes A + I_n \otimes B \otimes I_\ell + C \otimes I_m \otimes I_\ell \in \mathbb{R}^{\ell mn \times \ell mn}$, where $A \in \mathbb{R}^{\ell \times \ell}$, $B \in \mathbb{R}^{m \times m}$, $C \in \mathbb{R}^{n \times n}$. We focus on the shift-and-invert Lanczos method, which solves a shift-and-invert eigenvalue problem of $(T^T T - \tilde{\sigma}^2 I_{\ell mn})^{-1}$, where $\tilde{\sigma}$ is set to a scalar value close to the desired singular value. The desired singular value is computed by the maximum eigenvalue of the eigenvalue problem. This shift-and-invert Lanczos method needs to solve large-scale linear systems with the coefficient matrix $T^T T - \tilde{\sigma}^2 I_{\ell mn}$. The preconditioned conjugate gradient (PCG) method is applied since the direct methods cannot be applied due to the nonzero structure of the coefficient matrix. However, it is difficult in terms of memory requirements to simply implement the shift-and-invert Lanczos and the PCG methods since the size of $T$ grows rapidly by the sizes of $A$, $B$, and $C$. In this paper, we present the following two techniques: (1) efficient implementations of the shift-and-invert Lanczos method for the eigenvalue problem of $T^T T$ and the PCG method for $T^T T - \tilde{\sigma}^2 I_{\ell mn}$ using three-dimensional arrays (third-order tensors) and the $n$-mode products, and (2) preconditioning matrices of the PCG method based on the eigenvalue and the Schur decomposition of $T$. Finally, we show the effectiveness of the proposed methods through numerical experiments.

**Keywords:** tensor sum; singular value; shift-and-invert Lanczos method; preconditioned conjugate gradient method

**MSC:** 65F15; 65F08

## 1. Introduction

We consider computing an arbitrary singular value of a tensor sum:

$$T := I_n \otimes I_m \otimes A + I_n \otimes B \otimes I_\ell + C \otimes I_m \otimes I_\ell \in \mathbb{R}^{\ell mn \times \ell mn}, \tag{1}$$

where $A \in \mathbb{R}^{\ell \times \ell}$, $B \in \mathbb{R}^{m \times m}$, $C \in \mathbb{R}^{n \times n}$, $I_n$ is the $n \times n$ identity matrix, and the symbol "$\otimes$" denotes the Kronecker product. The tensor sum $T$ arises from a finite difference discretization of three-dimensional constant coefficient partial differential equations (PDE) defined as follows:

$$[-\boldsymbol{a} \cdot (\nabla * \nabla) + \boldsymbol{b} \cdot \nabla + c] u(x, y, z) = g(x, y, z) \text{ in } \Omega, \quad u(x, y, z) = 0 \text{ on } \partial\Omega, \tag{2}$$

where $\Omega = (0, 1) \times (0, 1) \times (0, 1)$, $\boldsymbol{a}, \boldsymbol{b} \in \mathbb{R}^3$, $c \in \mathbb{R}$, and the symbol "$*$" denotes element-wise products. If $\boldsymbol{a} = (1, 1, 1)$, then $\boldsymbol{a} \cdot (\nabla * \nabla) = \Delta$. Matrix $T$ tends to be too large even if $A, B$ and $C$ are not. Hence it is difficult to compute singular values of $T$ with regard to the memory requirement.

Previous studies [1,2] provided methods to compute the maximum and minimum singular values of $T$. By the previous studies, one can compute only the maximum and minimum singular values of $T$ without shift. On the other hand, one can compute arbitrary singular values of $T$ with the shift by this work. The previous studies are based on the Lanczos bidiagonalization method (see, e.g., [3]), which computes the maximum and minimum singular values of a matrix. For insights on Lanczos bidiagonalization method, see, e.g., [4–6]. The Lanczos bidiagonalization method for $T$ was implemented using tensors and their operations to reduce the memory requirement.

The Lanczos method with the shift-and-invert technique, see, e.g., [3], is widely known for computing an arbitrary eigenvalue $\lambda$ of a symmetric matrix $M \in \mathbb{R}^{n \times n}$. This method solves the shift-and-invert eigenvalue problem: $(M - \tilde{\sigma} I_n)^{-1} x = (\lambda - \tilde{\sigma})^{-1} x$, where $x$ is the eigenvector of $M$ corresponding to $\lambda$, and $\tilde{\sigma}$ is a shift point which is set to the nearby $\lambda$ ($\tilde{\sigma} \neq \lambda$). Since the eigenvalue problem has the eigenvalue $(\lambda - \tilde{\sigma})^{-1}$ as the maximum eigenvalue, the method is effective for computing the desired eigenvalue $\lambda$ near $\tilde{\sigma}$. For successful work using the shift-and-invert technique, see, e.g., [7–13].

Therefore, we obtain a computing method for an arbitrary singular value of $T$ based on the shift-and-invert Lanczos method. The method solves the following shift-and-invert eigenvalue problem: $(T^{\mathrm{T}} T - \tilde{\sigma}^2 I_{\ell mn})^{-1} x = (\sigma^2 - \tilde{\sigma}^2)^{-1} x$, where $\sigma$ is the desired singular value of $T$, $x$ is the corresponding right-singular vector, and $\tilde{\sigma}$ is close to $\sigma$ ($\tilde{\sigma} \neq \sigma$). This shift-and-invert Lanczos method requires the solution of large-scale linear systems with the coefficient matrix $T^{\mathrm{T}} T - \tilde{\sigma}^2 I_{\ell mn}$. Here, $T^{\mathrm{T}} T - \tilde{\sigma}^2 I_{\ell mn}$ can be a dense matrix whose number of elements is $O(n^6)$ even if $T$ is a sparse matrix whose number of elements is $O(n^4)$ when $A, B, C \in \mathbb{R}^{n \times n}$ are dense.

Since it is difficult regarding the memory requirement to apply the direct method, e.g., the Cholesky decomposition, which needs generating matrix $T^{\mathrm{T}} T - \tilde{\sigma}^2 I_{\ell mn}$, the pre-conditioned conjugate gradient (PCG) method, see, e.g., [14], is applied, even though it is difficult in terms of memory requirements to simply implement this shift-and-invert Lanczos method and the PCG method since the size of $T$ grows rapidly by the sizes of $A$, $B$, and $C$.

We propose the following two techniques in this paper: (1) Efficient implementations of the shift-and-invert Lanczos method for the eigenvalue problem of $T^{\mathrm{T}} T$ and the PCG method for $T^{\mathrm{T}} T - \tilde{\sigma}^2 I_{\ell mn}$ using three-dimensional arrays (third-order tensors) and the $n$-mode products, see, e.g., [15]. (2) Preconditioning matrices based on the eigenvalue decomposition and the Schur decomposition of $T$ for faster convergence of the PCG method. Finally, we show the effectiveness of the proposed method through numerical experiments.

## 2. Preliminaries of Tensor Operations

A tensor means a multidimensional array. Particularly, the third-order tensor $\mathcal{X} \in \mathbb{R}^{I \times J \times K}$ plays an important role. In the rest of this section, the definitions of some tensor operations are shown. For more details, see, e.g., [15].

Firstly, a summation, a subtraction, an inner product, and a norm for $\mathcal{X}, \mathcal{Y} \in \mathbb{R}^{I \times J \times K}$ are defined as follows:

$$(\mathcal{X} \pm \mathcal{Y})_{ijk} := \mathcal{X}_{ijk} \pm \mathcal{Y}_{ijk}, \quad (\mathcal{X}, \mathcal{Y}) := \sum_{i=1}^{I} \sum_{j=1}^{J} \sum_{k=1}^{K} \mathcal{X}_{ijk} \mathcal{Y}_{ijk}, \quad \|\mathcal{X}\| = \sqrt{(\mathcal{X}, \mathcal{X})},$$

where $\mathcal{X}_{ijk}$ denotes the $(i, j, k)$ element of $\mathcal{X}$. Secondly, the $n$-mode product of a tensor $\mathcal{X} \in \mathbb{R}^{I_1 \times I_2 \times \cdots \times I_N}$ and a matrix $M \in \mathbb{R}^{J \times I_n}$ is defined as

$$(\mathcal{X} \times_n M)_{i_1 \ldots i_{n-1} j i_{n+1} \ldots i_N} = \sum_{i_n=1}^{I_n} \mathcal{X}_{i_1 i_2 \ldots i_N} M_{p i_n},$$

where $n \in \{1, 2, \ldots N\}$, $i_k \in \{1, 2, \ldots, I_k\}$ for $k = 1, 2, \ldots N$, and $j \in \{1, 2, \ldots, J\}$. Finally, vec and vec$^{-1}$ operators are the following maps between a vector space $\mathbb{R}^{IJK}$ and a tensor

space $\mathbb{R}^{I \times J \times K}$: vec : $\mathbb{R}^{I \times J \times K} \to \mathbb{R}^{IJK}$ and vec$^{-1}$ : $\mathbb{R}^{IJK} \to \mathbb{R}^{I \times J \times K}$. vec operator can vectorize a tensor by combining all column vectors of the tensor into one long vector. Conversely, vec$^{-1}$ operator can reshape a tensor from one long vector.

### 3. Shift-and-Invert Lanczos Method for an Arbitrary Singular Value over Tensor Space

This section gives an algorithm for computing an arbitrary singular value of the tensor sum $T$. Let $\sigma$ and $x$ be a desired singular value of $T$ and the corresponding right singular vectors, respectively. Then, the eigenvalue problem of $T$ is written by $T^{\mathrm{T}}Tx = \sigma^2 x$. Here, introducing a shift $\tilde{\sigma} \approx \sigma$, the shift-and-invert eigenvalue problem is

$$(T^{\mathrm{T}}T - \tilde{\sigma}^2 I_{\ell mn})^{-1}x = \frac{1}{\sigma^2 - \tilde{\sigma}^2}x. \tag{3}$$

The shift-and-invert Lanczos method (see, e.g., [3]) computes the nearest singular value $\sigma$ based on Equation (3). Reconstructing this method over the $\ell \times m \times n$ tensor space, we obtain Algorithm 1 whose memory requirement is of $O(n^3)$ when $n = m = \ell$.

---

**Algorithm 1:** Shift-and-invert Lanczos method for an arbitrary singular value over tensor space

---

1: Choose an initial tensor $\mathcal{Q}_0 \in \mathbb{R}^{\ell \times m \times n}$;
2: $\mathcal{V} := \mathcal{Q}_0, \beta_0 := ||\mathcal{V}||$;
3: **for** $k = 1, 2, \ldots$, until convergence **do**
4:   $\mathcal{Q}_k := \mathcal{V} / \beta_{k-1}$;
5:   $\mathcal{V} := \mathrm{vec}^{-1}\left\{ \left(T^{\mathrm{T}}T - \tilde{\sigma}^2 I_{\ell mn}\right)^{-1}\mathrm{vec}(\mathcal{Q}_k) \right\}$;
      (Computed by Algorithms 3 or 4 in Section 4)
6:   $\mathcal{V} := \mathcal{V} - \beta_{k-1}\mathcal{Q}_{k-1}$;
7:   $\alpha_k := (\mathcal{Q}_k, \mathcal{V})$;
8:   $\mathcal{V} := \mathcal{V} - \alpha_k \mathcal{Q}_k$;
9:   $\beta_k := ||\mathcal{V}||$;
10: **end for**
11: Approximate singular value $\sigma = \sqrt{\tilde{\sigma}^2 + \dfrac{1}{\tilde{\lambda}^{(k)}}}$, where $\tilde{\lambda}^{(k)}$ is the maximum eigenvalue of $\tilde{T}_k$.

---

At step $k$, we have the following $\tilde{T}_k$ by Algorithm 1:

$$\tilde{T}_k := \begin{pmatrix} \alpha_1 & \beta_1 & & & \\ \beta_1 & \alpha_2 & \beta_2 & & \\ & \ddots & \ddots & \ddots & \\ & & \beta_{k-2} & \alpha_{k-1} & \beta_{k-1} \\ & & & \beta_{k-1} & \alpha_k \end{pmatrix} \in \mathbb{R}^{k \times k}.$$

To implement Algorithm 1, we need to iteratively solve the linear system

$$\mathcal{V} := \mathrm{vec}^{-1}\left\{ \left(T^{\mathrm{T}}T - \tilde{\sigma} I_{\ell mn}\right)^{-1}\mathrm{vec}(\mathcal{Q}_k) \right\}, \tag{4}$$

whose coefficient matrix is $\ell mn \times \ell mn$, that is, the memory requirement is $O(n^6)$ when $n = m = \ell$. Here, the convergence rate of the shift and invert Lanczos method depends on the ratio of gaps between the maximum, the second maximum, and the minimum singular values $\sigma_1, \sigma_2, \sigma_m$ of $(T^{\mathrm{T}}T - \tilde{\sigma}^2 I_{\ell mn})^{-1}$ as follows: $(\sigma_1^2 - \sigma_2^2)/(\sigma_1^2 - \sigma_m^2)$.

In the next section, we consider solving the linear systems with memory requirement of $O(n^3)$ when $n = m = \ell$.

## 4. Preconditioned Conjugate Gradient (PCG) Method over Tensor Space

This section provides an efficient solver of Equation (4) using tensors. This linear system is rewritten by $v = \left(T^{\mathrm{T}}T - \tilde{\sigma}^2 I_{\ell mn}\right)^{-1} q_k$, where $v := \mathrm{vec}(\mathcal{V})$ and $q_k := \mathrm{vec}(\mathcal{Q}_k)$. Then we solve $\left(T^{\mathrm{T}}T - \tilde{\sigma}^2 I_{\ell mn}\right) v = q_k$, where $v$ and $q_k$ are unknown and known vectors. Since the coefficient matrix is symmetric positive definite, we can use the preconditioned conjugate gradient method (PCG method, see, e.g., [14]), which is one of the widely used solvers. However, it is difficult to simply apply the method due to the complex nonzero structure of the coefficient matrix $T^{\mathrm{T}}T - \tilde{\sigma}^2 I_{\ell mn}$. For applying the PCG method, we consider transforming the linear system $\left(T^{\mathrm{T}}T - \tilde{\sigma}^2 I_{\ell mn}\right) v = q_k$ by the eigendecomposition and the complex Schur decomposition as shown in the next subsections.

### 4.1. PCG Method by the Eigendecomposition

Firstly, $T$ is decomposed into $T := XDX^{-1}$, where $X$ and $D$ are a matrix whose column vectors are eigenvectors and a diagonal matrix with eigenvalues, respectively. Then, it follows that

$$\left(T^{\mathrm{T}}T - \tilde{\sigma}^2 I_{\ell mn}\right) v = q_k \Leftrightarrow \left((XDX^{-1})^{\mathrm{H}}(XDX^{-1}) - \tilde{\sigma}^2 I_{\ell mn}\right) v = q_k$$
$$\Leftrightarrow \left(\overline{D}X^{\mathrm{H}}XD - \tilde{\sigma}^2 X^{\mathrm{H}}X\right)\left(X^{-1}v\right) = X^{\mathrm{H}}q_k,$$

where $\overline{D}$ is the complex conjugate of $D$. We rewrite the above linear system into $\tilde{A}\tilde{y} = \tilde{b}$, where $\tilde{A} := \overline{D}X^{\mathrm{H}}XD - \tilde{\sigma}^2 X^{\mathrm{H}}X$, $\tilde{y} := X^{-1}v$, and $\tilde{b} := X^{\mathrm{H}}q_k$. Here, $X$ is easily computed by small matrices $X_A$, $X_B$, and $X_C$ whose column vectors are eigenvectors of $A$, $B$, and $C$ as follows: $X = X_C \otimes X_B \otimes X_A$. Moreover, eigenvalues of $T$ in $D$ are obtained by summations of each eigenvalue of $A$, $B$, and $C$.

The PCG method for solving $\tilde{A}\tilde{y} = \tilde{b}$ is shown in Algorithm 2. Since this algorithm computes $\tilde{y}$, we need to compute $v = X\tilde{y}$. Section 4.1.1 proposes a preconditioning matrix and Section 4.1.2 provides efficient computations using tensors.

---

**Algorithm 2:** PCG method over vector space for $\tilde{A}\tilde{y} = \tilde{b}$

---

1: Choose an initial vector $x_0 \in \mathbb{R}^{\ell mn}$ and $p_0 = \mathbf{0} \in \mathbb{R}^{\ell mn}$, and an initial scalar $\beta_0 = 0$;
2: $r_0 = \tilde{b} - \tilde{A}x_0$;
3: $z_0 = M^{-1}r_0$;
4: **for** $k' = 1, 2, \ldots$, until convergence **do**
5:     $p_{k'} = z_{k'-1} + \beta_{k'-1}p_{k'-1}$;
6:     $\hat{p}_{k'} = \tilde{A}p_{k'}$;
7:     $\alpha_{k'} = (z_{k'-1}, r_{k'-1})/(p_{k'-1}, \hat{p}_{k'})$;
8:     $x_{k'} = x_{k'-1} + \alpha_{k'}p_{k'}$;
9:     $r_{k'} = r_{k'-1} - \alpha_{k'}\hat{p}_{k'}$;
10:    $z_{k'} = M^{-1}r_{k'}$;
11:    $\beta_{k'} = (z_{k'}, r_{k'})/(z_{k'-1}, r_{k'-1})$;
12: **end for**
13: Obtain an approximate solution $\tilde{y} \approx x_{k'}$;

---

#### 4.1.1. Preconditioning Matrix

Algorithm 2 solves

$$\left(M^{-1}\tilde{A}M^{-\mathrm{H}}\right)\left(M^{\mathrm{H}}\tilde{y}\right) = M^{-1}\tilde{b},$$

where $M \in \mathbb{R}^{\ell mn \times \ell mn}$ is a preconditioning matrix. $M$ must satisfy the following two conditions: (1) a condition number of $M^{-1}\tilde{A}$ is close to 1; (2) the matrix-vector multiplication of $M^{-1}$ is easily computed.

Therefore, we propose a preconditioning matrix based on the eigendecomposition of $T$

$$M := \overline{D}D - \tilde{\sigma}^2 I_{\ell mn}. \tag{5}$$

Since $M$ is the diagonal matrix, the second condition of the preconditioning matrix is satisfied. Moreover, if $T$ is symmetric, $X$ is the unitary matrix, that is, $X^{\mathrm{H}}X = I_{\ell mn}$. In the case of the symmetric matrix $T$, we obtain $M = \tilde{A}$. Namely, the proposed matrix satisfies the first conditions when $T$ is symmetric. So, even if $T$ is not exactly symmetric, if $T$ is almost symmetric, then we can expect the preconditioning matrix $M$ to be effective.

### 4.1.2. Efficient Implementation of Algorithm 2 by the Eigendecomposition

Similarly to obtaining Algorithm 1, to improve an implementation of Algorithm 2, we reconstruct $\ell mn$ dimensional vectors into $\ell \times m \times n$ tensors via $\mathrm{vec}^{-1}$ operator as follows: $\boldsymbol{\mathcal{X}}_{k'} := \mathrm{vec}^{-1}(\boldsymbol{x}_{k'})$, $\boldsymbol{\mathcal{R}}_{k'} := \mathrm{vec}^{-1}(\boldsymbol{r}_{k'})$, $\boldsymbol{\mathcal{P}}_{k'} := \mathrm{vec}^{-1}(\boldsymbol{p}_{k'})$, $\boldsymbol{\mathcal{Z}}_{k'} := \mathrm{vec}^{-1}(\boldsymbol{z}_{k'})$, and $\hat{\boldsymbol{\mathcal{P}}}_{k'} := \mathrm{vec}^{-1}(\hat{\boldsymbol{p}}_{k'})$. Most computations of vectors are simply transformed into computations of tensors because of the linearity of $\mathrm{vec}^{-1}$ operator.

In the rest of this section, we show the computations of $\mathrm{vec}^{-1}\big(\tilde{A}\mathrm{vec}(\boldsymbol{\mathcal{P}}_{k'})\big)$ and $\mathrm{vec}^{-1}\big(M^{-1}\mathrm{vec}(\boldsymbol{\mathcal{R}}_{k'})\big)$, which are required in the PCG method, using the 1, 2, and 3-mode products for tensors and the definition of $T$. First, from the definitions of $\tilde{A}$ and $X$, $\mathrm{vec}^{-1}\big(\tilde{A}\mathrm{vec}(\boldsymbol{\mathcal{P}}_{k'})\big) = \mathrm{vec}^{-1}(\overline{D}X^{\mathrm{H}}XD\mathrm{vec}(\boldsymbol{\mathcal{P}}_{k'})) - \tilde{\sigma}^2\mathrm{vec}^{-1}(X^{\mathrm{H}}X\mathrm{vec}(\boldsymbol{\mathcal{P}}_{k'}))$ holds. Let $\boldsymbol{\mathcal{D}} = \mathrm{vec}^{-1}(\mathrm{diag}(D))$, where $\mathrm{diag}(D)$ returns an $\ell mn$-dimensional column vector with diagonals of $D$. Then, $\boldsymbol{\mathcal{D}}_{ijk} := \lambda_i^{(A)} + \lambda_j^{(B)} + \lambda_k^{(C)}$, where $\lambda_i^{(A)}, \lambda_j^{(B)}$, and $\lambda_k^{(C)}$ denote the eigenvalues of $A, B$, and $C$. Note that $(\mathrm{vec}^{-1}(D\mathrm{vec}(\boldsymbol{\mathcal{P}}_{k'})))_{ijk} = \boldsymbol{\mathcal{D}}_{ijk}(\boldsymbol{\mathcal{P}}_{k'})_{ijk}$ since we compute $(D\boldsymbol{p}_{k'})_i = D_{ii}(\boldsymbol{p}_{k'})_i$ for $i = 1, 2, \ldots, \ell mn$. Using the relation between the Kronecker product and the mode products via $\mathrm{vec}^{-1}$ operator, we compute

$$
\begin{aligned}
&\mathrm{vec}^{-1}\big(\tilde{A}\mathrm{vec}(\boldsymbol{\mathcal{P}})\big) \\
&= \overline{\boldsymbol{\mathcal{D}}} * \Big\{ (\boldsymbol{\mathcal{D}} * \boldsymbol{\mathcal{P}}_{k'}) \times_1 X_A^{\mathrm{H}}X_A + (\boldsymbol{\mathcal{D}} * \boldsymbol{\mathcal{P}}_{k'}) \times_2 X_B^{\mathrm{H}}X_B + (\boldsymbol{\mathcal{D}} * \boldsymbol{\mathcal{P}}_{k'}) \times_3 X_C^{\mathrm{H}}X_C \Big\} \\
&\quad - \tilde{\sigma}^2 \Big( \boldsymbol{\mathcal{P}}_{k'} \times_1 X_A^{\mathrm{H}}X_A + \boldsymbol{\mathcal{P}}_{k'} \times_2 X_B^{\mathrm{H}}X_B + \boldsymbol{\mathcal{P}}_{k'} \times_3 X_C^{\mathrm{H}}X_C \Big),
\end{aligned} \tag{6}
$$

where "$*$" denotes elementwise product.

Next, from the definition of the diagonal matrix $M$ in Equation (5), we easily obtain

$$\left( M^{-1} \right)_{ii} = \frac{1}{(\overline{D})_{ii}(D)_{ii} - \tilde{\sigma}^2}, \qquad i = 1, 2, \ldots, \ell mn.$$

Here, let $\boldsymbol{\mathcal{M}} = \mathrm{vec}^{-1}(\mathrm{diag}(M^{-1}))$. Then it follows that $\boldsymbol{\mathcal{M}}_{ijk} = 1/(\overline{\boldsymbol{\mathcal{D}}}_{ijk}\boldsymbol{\mathcal{D}}_{ijk} - \tilde{\sigma}^2)$. $\mathrm{vec}^{-1}\big(M^{-1}\mathrm{vec}(\boldsymbol{\mathcal{R}}_{k'})\big)$ is computed by

$$\mathrm{vec}^{-1}\Big( M^{-1}\mathrm{vec}(\boldsymbol{\mathcal{R}}_{k'}) \Big) = \boldsymbol{\mathcal{M}} * \boldsymbol{\mathcal{R}}_{k'}. \tag{7}$$

As shown in Algorithm 3, the PCG method can be implemented using the preconditioning matrix $M$ and the aforementioned computations, where the linear system $\tilde{A}\tilde{\boldsymbol{y}} = \tilde{\boldsymbol{b}}$ is transformed into $\tilde{A}\,\mathrm{vec}(\tilde{\boldsymbol{\mathcal{Y}}}) = \mathrm{vec}(\tilde{\boldsymbol{\mathcal{B}}})$, where $\mathrm{vec}(\tilde{\boldsymbol{\mathcal{B}}}) := \tilde{\boldsymbol{b}} = \mathrm{vec}(\boldsymbol{\mathcal{Q}}_k \times_1 X_A^{\mathrm{H}} + \boldsymbol{\mathcal{Q}}_k \times_2 X_B^{\mathrm{H}} + \boldsymbol{\mathcal{Q}}_k \times_3 X_C^{\mathrm{H}})$ and $\mathrm{vec}(\tilde{\boldsymbol{\mathcal{Y}}}) := \tilde{\boldsymbol{y}}$. Algorithm 3 requires only small matrices $A, B$, and $C$ and $\ell \times m \times n$ tensors $\boldsymbol{\mathcal{X}}_{k'}, \boldsymbol{\mathcal{R}}_{k'}, \boldsymbol{\mathcal{P}}_{k'}$, and $\boldsymbol{\mathcal{Z}}_{k'}$. Therefore the memory requirement is of $O(n^3)$ in the case of $n = m = \ell$.

### 4.2. PCG Method by the Schur Decomposition

Firstly, the Schur decomposition of $T$ is $T := QRQ^{\mathrm{H}}$, where $R$ and $Q$ are upper triangular and unitary matrices, respectively. Then,

$$\left(T^{\mathrm{T}}T - \tilde{\sigma}^2 I_{\ell mn}\right)\boldsymbol{v} = \boldsymbol{q}_k \Leftrightarrow \left((QRQ^{\mathrm{H}})^{\mathrm{H}}(QRQ^{\mathrm{H}}) - \tilde{\sigma}^2 I_{\ell mn}\right)\boldsymbol{v} = \boldsymbol{q}_k$$

$$\Leftrightarrow \left(R^{\mathrm{H}}R - \tilde{\sigma}^2 I_{\ell mn}\right)(Q^{\mathrm{H}}\boldsymbol{v}) = Q^{\mathrm{H}}\boldsymbol{q}_k.$$

This linear system denotes $\tilde{A}\tilde{\boldsymbol{y}} = \tilde{\boldsymbol{b}}$, where $\tilde{A} := R^{\mathrm{H}}R - \tilde{\sigma}^2 I_{\ell mn}$, $\tilde{\boldsymbol{y}} := Q^{\mathrm{H}}\boldsymbol{v}$, and $\tilde{\boldsymbol{b}} := Q^{\mathrm{H}}\boldsymbol{q}_k$. The PCG method for $\tilde{A}\tilde{\boldsymbol{y}} = \tilde{\boldsymbol{b}}$ is shown in Algorithm 2. $R$ and $Q$ are obtained from the complex Schur decomposition of $A, B,$ and $C$ as follows: $R = I_n \otimes I_m \otimes R_A + I_n \otimes R_B \otimes I_\ell + R_C \otimes I_m \otimes I_\ell$ and $Q = Q_C \otimes Q_B \otimes Q_A$ from the definition of $T$, where $A = Q_A R_A Q_A^{\mathrm{H}}$, $B = Q_B R_B Q_B^{\mathrm{H}}$, and $C = Q_C R_C Q_C^{\mathrm{H}}$ by the Schur decomposition of $A, B,$ and $C$.

---

**Algorithm 3:** PCG method over tensor space for the 5-th line of Algorithm 1 [Proposed inner algorithm using the eigendecomposition]

---

1: Choose an initial tensor $\boldsymbol{\mathcal{X}}_0 \in \mathbb{R}^{\ell \times m \times n}$ and $\boldsymbol{\mathcal{P}}_0 = O_{\ell \times m \times n}$, and an initial scalar $\beta_0 = 0$;
2: $\boldsymbol{\mathcal{R}}_0 = \left(\boldsymbol{\mathcal{Q}}_k \times_1 X_A^{\mathrm{H}} + \boldsymbol{\mathcal{Q}}_k \times_2 X_B^{\mathrm{H}} + \boldsymbol{\mathcal{Q}}_k \times_3 X_C^{\mathrm{H}}\right)$
　　　$- \left[\overline{\boldsymbol{\mathcal{D}}} * \left\{(\boldsymbol{\mathcal{D}} * \boldsymbol{\mathcal{X}}_0) \times_1 X_A^{\mathrm{H}}X_A + (\boldsymbol{\mathcal{D}} * \boldsymbol{\mathcal{X}}_0) \times_2 X_B^{\mathrm{H}}X_B + (\boldsymbol{\mathcal{D}} * \boldsymbol{\mathcal{X}}_0) \times_3 X_C^{\mathrm{H}}X_C\right\}\right.$
　　　$\left.- \tilde{\sigma}^2(\boldsymbol{\mathcal{X}}_0 \times_1 X_A^{\mathrm{H}}X_A + \boldsymbol{\mathcal{X}}_0 \times_2 X_B^{\mathrm{H}}X_B + \boldsymbol{\mathcal{X}}_0 \times_3 X_C^{\mathrm{H}}X_C)\right];$
3: $\boldsymbol{\mathcal{Z}}_0 = \boldsymbol{\mathcal{M}} * \boldsymbol{\mathcal{R}}_0;$
4: **for** $k' = 1, 2, \ldots,$ until convergence **do**
5: 　$\boldsymbol{\mathcal{P}}_{k'} = \boldsymbol{\mathcal{Z}}_{k'-1} + \beta_{k'-1}\boldsymbol{\mathcal{P}}_{k'-1};$
6: 　$\hat{\boldsymbol{\mathcal{P}}}_{k'} = \overline{\boldsymbol{\mathcal{D}}} * \left\{(\boldsymbol{\mathcal{D}} * \boldsymbol{\mathcal{P}}_{k'}) \times_1 X_A^{\mathrm{H}}X_A + (\boldsymbol{\mathcal{D}} * \boldsymbol{\mathcal{P}}_{k'}) \times_2 X_B^{\mathrm{H}}X_B + (\boldsymbol{\mathcal{D}} * \boldsymbol{\mathcal{P}}_{k'}) \times_3 X_C^{\mathrm{H}}X_C\right\}$
　　　$- \tilde{\sigma}^2\left(\boldsymbol{\mathcal{P}}_{k'} \times_1 X_A^{\mathrm{H}}X_A + \boldsymbol{\mathcal{P}}_{k'} \times_2 X_B^{\mathrm{H}}X_B + \boldsymbol{\mathcal{P}}_{k'} \times_3 X_C^{\mathrm{H}}X_C\right);$
7: 　$\alpha_{k'} = (\boldsymbol{\mathcal{Z}}_{k'-1}, \boldsymbol{\mathcal{R}}_{k'-1}) / \left(\boldsymbol{\mathcal{P}}_{k'-1}, \hat{\boldsymbol{\mathcal{P}}}_{k'}\right);$
8: 　$\boldsymbol{\mathcal{X}}_{k'} = \boldsymbol{\mathcal{X}}_{k'-1} + \alpha_{k'}\boldsymbol{\mathcal{P}}_{k'};$
9: 　$\boldsymbol{\mathcal{R}}_{k'} = \boldsymbol{\mathcal{R}}_{k'-1} - \alpha_{k'}\hat{\boldsymbol{\mathcal{P}}}_{k'};$
10: 　$\boldsymbol{\mathcal{Z}}_{k'} = \boldsymbol{\mathcal{M}} * \boldsymbol{\mathcal{R}}_{k'-1};$
11: 　$\beta_{k'} = (\boldsymbol{\mathcal{Z}}_{k'}, \boldsymbol{\mathcal{R}}_{k'}) / (\boldsymbol{\mathcal{Z}}_{k'-1}, \boldsymbol{\mathcal{R}}_{k'-1});$
12: **end for**
13: Obtain an approximate solution $\tilde{\boldsymbol{\mathcal{Y}}} \approx \boldsymbol{\mathcal{X}}_{k'};$
14: $\boldsymbol{\mathcal{V}} = \tilde{\boldsymbol{\mathcal{Y}}} \times_1 X_A + \tilde{\boldsymbol{\mathcal{Y}}} \times_2 X_B + \tilde{\boldsymbol{\mathcal{Y}}} \times_3 X_C;$

---

#### 4.2.1. Preconditioning Matrix

A preconditioning matrix for $\tilde{A}\tilde{\boldsymbol{y}} = \tilde{\boldsymbol{b}}$ satisfies the conditions in Section 4.1.1. Therefore, we propose the preconditioning matrix based on the Schur decomposition

$$M := \overline{D}_R D_R - \tilde{\sigma}^2 I_{\ell mn},$$

where $D_R$ is a diagonal matrix with diagonals of $R$. Since $M$ is also the diagonal matrix, the above second conditions are satisfied. Moreover, if $T$ is symmetric, $R$ is a diagonal matrix, that is, $R = D_R$. Therefore $M = \tilde{A}$ in the case of the symmetric matrix $T$. From this, we expect that the preconditioning matrix $M$ is effective if $T$ is not symmetric but almost symmetric.

#### 4.2.2. Efficient Implementation of Algorithm 2 by the Schur Decomposition

We show the computations of $\text{vec}^{-1}\left(\tilde{A}\text{vec}(\boldsymbol{\mathcal{P}}_{k'})\right)$ and $\text{vec}^{-1}\left(M^{-1}\text{vec}(\boldsymbol{\mathcal{R}}_{k'})\right)$ for the PCG method over tensor space using the 1, 2, and 3-mode products for tensors and the

definition of $T$. First, from the definitions of $\tilde{A}$ and $R$, we have $\text{vec}^{-1}(\tilde{A}\text{vec}(\boldsymbol{\mathcal{P}}_{k'})) = \text{vec}^{-1}(R^{\text{H}}(R\text{vec}(\boldsymbol{\mathcal{P}}_{k'})) - \tilde{\sigma}^2\text{vec}(\boldsymbol{\mathcal{P}}_{k'}))$. Therefore,

$$\text{vec}^{-1}(\tilde{A}\text{vec}(\boldsymbol{\mathcal{P}}_{k'})) = \boldsymbol{\mathcal{P}}_{k'} \times_1 R_A^{\text{H}} R_A + \boldsymbol{\mathcal{P}}_{k'} \times_2 R_B^{\text{H}} R_B + \boldsymbol{\mathcal{P}}_{k'} \times_3 R_C^{\text{H}} R_C - \tilde{\sigma}^2 \boldsymbol{\mathcal{P}}_{k'}.$$

Next, from $M = \overline{D}_R D_R - \tilde{\sigma}^2 I_{\ell mn}$, we easily obtain

$$\left(M^{-1}\right)_{ii} = \frac{1}{(\overline{D}_R)_{ii}(D_R)_{ii} - \tilde{\sigma}^2}, \qquad i = 1, 2, \ldots, \ell mn.$$

Similarly to Section 4.1.2, let $\boldsymbol{\mathcal{D}} = \text{vec}^{-1}(\text{diag}(D_R))$ and $\boldsymbol{\mathcal{M}} = \text{vec}^{-1}(\text{diag}(M^{-1}))$. Then, we have $\boldsymbol{\mathcal{M}}_{ijk} = 1/(\overline{\boldsymbol{\mathcal{D}}}_{ijk}\boldsymbol{\mathcal{D}}_{ijk} - \tilde{\sigma}^2)$, where $\boldsymbol{\mathcal{D}}_{ijk} = (R_A)_{ii} + (R_B)_{jj} + (R_C)_{kk}$. $\text{vec}^{-1}(M^{-1}\text{vec}(\boldsymbol{\mathcal{R}}_{k'}))$ is computed by (7).

As shown in Algorithm 4, the PCG method can be implemented using the preconditioning matrix $M$ and the aforementioned computations, where the linear system $\tilde{A}\tilde{\boldsymbol{y}} = \tilde{\boldsymbol{b}}$ is transformed into $\tilde{A}\,\text{vec}(\tilde{\boldsymbol{\mathcal{Y}}}) = \text{vec}(\tilde{\boldsymbol{\mathcal{B}}})$, where $\text{vec}(\tilde{\boldsymbol{\mathcal{B}}}) := \tilde{\boldsymbol{b}} = \text{vec}(\boldsymbol{\mathcal{Q}}_k \times_1 Q_A + \boldsymbol{\mathcal{Q}}_k \times_2 Q_B + \boldsymbol{\mathcal{Q}}_k \times_3 Q_C)$ and $\text{vec}(\tilde{\boldsymbol{\mathcal{Y}}}) := \tilde{\boldsymbol{y}}$. Algorithm 4 just requires small matrices $A$, $B$, and $C$ and $\ell \times m \times n$ tensors $\boldsymbol{\mathcal{X}}_{k'}, \boldsymbol{\mathcal{R}}_{k'}, \boldsymbol{\mathcal{P}}_{k'}$, and $\boldsymbol{\mathcal{Z}}_{k'}$, namely, do not require large matrix $T$. Therefore the memory requirement is of $O(n^3)$ in the case of $n = m = \ell$.

---

**Algorithm 4:** PCG method over tensor space for the 5-th line of Algorithm 1 [Proposed inner algorithm using the Schur decomposition]

---

1: Choose an initial tensor $\boldsymbol{\mathcal{X}}_0 \in \mathbb{R}^{\ell \times m \times n}$ and $\boldsymbol{\mathcal{P}}_0 = O_{\ell \times m \times n}$, and an initial scalar $\beta_0 = 0$;
2: $\boldsymbol{\mathcal{R}}_0 = (\boldsymbol{\mathcal{Q}}_k \times_1 Q_A + \boldsymbol{\mathcal{Q}}_k \times_2 Q_B + \boldsymbol{\mathcal{Q}}_k \times_3 Q_C)$
$\qquad - (\boldsymbol{\mathcal{X}}_0 \times_1 R_A^{\text{H}} R_A + \boldsymbol{\mathcal{X}}_0 \times_2 R_B^{\text{H}} R_B + \boldsymbol{\mathcal{X}}_0 \times_3 R_C^{\text{H}} R_C - \tilde{\sigma}^2 \boldsymbol{\mathcal{X}}_0)$;
3: $\boldsymbol{\mathcal{Z}}_0 = \boldsymbol{\mathcal{M}} * \boldsymbol{\mathcal{R}}_0$;
4: **for** $k' = 1, 2, \ldots$, until convergence **do**
5: $\quad \boldsymbol{\mathcal{P}}_{k'} = \boldsymbol{\mathcal{Z}}_{k'-1} + \beta_{k'-1} \boldsymbol{\mathcal{P}}_{k'-1}$;
6: $\quad \hat{\boldsymbol{\mathcal{P}}}_{k'} = \boldsymbol{\mathcal{P}}_{k'} \times_1 R_A^{\text{H}} R_A + \boldsymbol{\mathcal{P}}_{k'} \times_2 R_B^{\text{H}} R_B + \boldsymbol{\mathcal{P}}_{k'} \times_3 R_C^{\text{H}} R_C - \tilde{\sigma}^2 \boldsymbol{\mathcal{P}}_{k'}$;
7: $\quad \alpha_{k'} = (\boldsymbol{\mathcal{Z}}_{k'-1}, \boldsymbol{\mathcal{R}}_{k'-1})/(\boldsymbol{\mathcal{P}}_{k'-1}, \hat{\boldsymbol{\mathcal{P}}}_{k'})$;
8: $\quad \boldsymbol{\mathcal{X}}_{k'} = \boldsymbol{\mathcal{X}}_{k'-1} + \alpha_{k'} \boldsymbol{\mathcal{P}}_{k'}$;
9: $\quad \boldsymbol{\mathcal{R}}_{k'} = \boldsymbol{\mathcal{R}}_{k'-1} - \alpha_{k'} \hat{\boldsymbol{\mathcal{P}}}_{k'}$;
10: $\quad \boldsymbol{\mathcal{Z}}_{k'} = \boldsymbol{\mathcal{M}} * \boldsymbol{\mathcal{R}}_{k'}$;
11: $\quad \beta_{k'} = (\boldsymbol{\mathcal{Z}}_{k'}, \boldsymbol{\mathcal{R}}_{k'})/(\boldsymbol{\mathcal{Z}}_{k'-1}, \boldsymbol{\mathcal{R}}_{k'-1})$;
12: **end for**
13: Obtain an approximate solution $\tilde{\boldsymbol{\mathcal{Y}}} \approx \boldsymbol{\mathcal{X}}_{k'}$;
14: $\boldsymbol{\mathcal{V}} = \tilde{\boldsymbol{\mathcal{Y}}} \times_1 Q_A + \tilde{\boldsymbol{\mathcal{Y}}} \times_2 Q_B + \tilde{\boldsymbol{\mathcal{Y}}} \times_3 Q_C$;

---

## 5. Numerical Experiments

This section provides results of numerical experiments using Algorithm 1 with Algorithm 3 and Algorithm 1 with Algorithm 4. There are the two purposes of this experiments: (1) to confirm convergence to the singular value of $T$ nearest to the shift by Algorithm 1, and (2) to confirm the effectiveness of the proposed precondition matrix in Algorithms 3 and 4. For comparison, the results using Algorithms 3 and 4 in the case of $M = I$ are also given as the results by the CG method. All the initial guesses of Algorithms 1, 3, and 4 are tensors with random numbers. The stopping criteria used in Algorithm 1 was $\beta_k \|e_k^{\text{T}} s_{\text{MAX}}^k\| < 10^{-8}$, where $s_{\text{MAX}}^k$ is the eigenvector corresponding to the maximum eigenvalue of $\tilde{T}_k$ and $e_k$ denotes the $k$-th canonical basis for $k$ dimensional vector space. Algorithms 3 and 4 were stopped when either the relative residual $\|\boldsymbol{\mathcal{R}}_{k'}\|/\|\tilde{\boldsymbol{\mathcal{B}}}\| < 10^{-12}$ or the maximum number of iterations $k' > 20{,}000$ were satisfied.

All computations were carried out using MATLAB R2021a version on a workstation with Xeon processor 3.7 GHz and 128 GB of RAM.

In the following subsection, we show the results computing the 5-th maximum, median, and 5-th minimum singular values $\sigma$ of the test matrices $T$. For all the cases, for the first purpose, we set the shift value in Algorithm 1 as

$$\tilde{\sigma} = \sigma - 10^{-2}, \tag{8}$$

where $\tilde{\sigma}$'s and $\sigma$'s are the perturbed singular values of $T$ and the aforementioned singular values computed by the svd function in MATLAB, respectively.

Test matrices $T$ in Equation (1) are obtained from a seven-point central difference discretization of the PDE (2) in over an $(n+1) \times (n+1) \times (n+1)$ grid. The test matrices $T$ in Equation (1), whose size is $n^3 \times n^3$, are generated from

$$A = B = C, \quad A := \frac{1}{h^2}aM_1 + \frac{1}{2h}bM_2 + \frac{1}{3}cI_n, \tag{9}$$

where $h = 1/(n+1)$, $M_1$ and $M_2$ are symmetric and skew-symmetric matrices given below.

$$M_1 = \begin{pmatrix} -2 & 1 & & & \\ 1 & -2 & 1 & & \\ & \ddots & \ddots & \ddots & \\ & & 1 & -2 & 1 \\ & & & 1 & -2 \end{pmatrix} \in \mathbb{R}^{n \times n}, M_2 = \begin{pmatrix} 0 & 1 & & & \\ -1 & 0 & 1 & & \\ & \ddots & \ddots & \ddots & \\ & & -1 & 0 & 1 \\ & & & -1 & 0 \end{pmatrix} \in \mathbb{R}^{n \times n}.$$

*Numerical Results*

In all tables, the number of iterations of the shift-and-invert Lanczos method ("the Lanczos method" hereafter) and the average of the number of iterations of the CG or PCG method based on the eigendecomposition or the Schur decomposition are summarized. "Not converged" denotes Algorithm 3 or 4 did not converge.

We show the first results in the case of almost symmetric matrix with $a = c = 1$ and $b = 0.01$ in Equation (9) for the shift (8). From Tables 1–3, the numbers of iterations of Lanczos methods using any inner algorithms were almost the same. Focusing on the effectiveness of the proposed preconditioning matrix $M$, the numbers of iterations of both PCG methods were less than 19 regardless of the size of $T$. On the other hand, the numbers of iterations of both CG methods linearly increased depending on the size of $T$. From these facts, the preconditioning matrix $M$ is effective in the case of almost symmetric matrix $T$. Moreover, the number of iterations of the shift and invert Lanczos method for the median singular value is larger than the number for other singular values since the distance between the maximum and second maximum singular values of $(T^{\mathsf{T}}T - \tilde{\sigma}^2 I_{\ell mn})^{-1}$ for the median singular value of $T$ is closer than the cases of other singular values.

Here, the running time of Table 1 is summarized in Table 4. All the running time by the PCG method were less than the time by the CG method. Moreover, the running time by the PCG methods of Algorithms 3 and 4 were similar since the computational complexities of these algorithms are similar. Thus, the running time is strongly correlated with the number of iterations of Algorithms 3 and 4.

In addition, convergence histories of $n = 15$ in Tables 1 and 2 are shown in Figures 1 and 2. Figure 2 displays that the relative residual norms unsteadily decreased when the number of iterations of the shift and invert Lanczos method is not small.

**Table 1.** Number of iterations of the Lanczos method and the average of numbers of iterations of the CG/PCG method in the case of the 5-th max. singular value of almost symmetric matrix $T$ with $a = c = 1$ and $b = 0.01$ in (9) for the shift (8).

|        | Method | Algorithms 1 and 3 (by Eigendecompn.) | | | | Algorithms 1 and 4 (by Schur Decompn.) | | | |
|--------|--------|---------|-------|---------|-------|---------|-------|---------|-------|
|        |        | Lanczos | CG    | Lanczos | PCG   | Lanczos | CG    | Lanczos | PCG   |
|        | 5      | 4       | 43.0  | 4       | 16.0  | 4       | 35.8  | 4       | 15.0  |
|        | 10     | 4       | 90.0  | 4       | 17.0  | 4       | 86.5  | 4       | 15.0  |
| $n$    | 15     | 3       | 134.7 | 3       | 17.0  | 3       | 128.0 | 3       | 17.0  |
|        | 20     | 4       | 180.0 | 4       | 17.0  | 4       | 169.3 | 4       | 17.0  |
|        | 25     | 3       | 225.7 | 3       | 17.0  | 3       | 211.0 | 3       | 17.0  |
|        | 30     | 3       | 273.0 | 3       | 17.0  | 3       | 252.3 | 3       | 17.0  |

**Table 2.** Number of iterations of the Lanczos method and the average of numbers of iterations of the CG/PCG method in the case of the median of singular value of almost symmetric matrix $T$ with $a = c = 1$ and $b = 0.01$ in (9) for the shift (8).

|        | Method | Algorithms 1 and 3 (by Eigendecompn.) | | | | Algorithms 1 and 4 (by Schur Decompn.) | | | |
|--------|--------|-----------------|--------|---------|-------|-----------------|--------|---------|-------|
|        |        | Lanczos         | CG     | Lanczos | PCG   | Lanczos         | CG     | Lanczos | PCG   |
|        | 5      | 15              | 139.3  | 15      | 13.0  | 15              | 109.1  | 15      | 15.0  |
|        | 10     | 7               | 1081.0 | 7       | 13.0  | 7               | 943.7  | 7       | 15.0  |
| $n$    | 15     | 41              | 5201.6 | 39      | 15.0  | 39              | 4858.1 | 41      | 17.0  |
|        | 20     | 13              | 8339.5 | 4       | 16.0  | 4               | 6847.3 | 4       | 15.0  |
|        | 25     | (Not converged.)|        | 48      | 17.0  | (Not converged.)|        | 48      | 17.0  |
|        | 30     | (Not converged.)|        | 8       | 19.0  | (Not converged.)|        | 8       | 15.0  |

**Table 3.** Number of iterations of the Lanczos method and the average of numbers of iterations of the CG/PCG method in the case of the 5-th min. singular value of almost symmetric matrix $T$ with $a = c = 1$ and $b = 0.01$ in (9) for the shift (8).

|        | Method | Algorithms 1 and 3 (by Eigendecompn.) | | | | Algorithms 1 and 4 (by Schur Decompn.) | | | |
|--------|--------|---------|--------|---------|-------|---------|--------|---------|-------|
|        |        | Lanczos | CG     | Lanczos | PCG   | Lanczos | CG     | Lanczos | PCG   |
|        | 5      | 3       | 124.7  | 3       | 13.0  | 3       | 97.9   | 3       | 14.6  |
|        | 10     | 3       | 748.1  | 3       | 13.0  | 3       | 654.9  | 3       | 14.1  |
| $n$    | 15     | 3       | 4872.5 | 3       | 14.9  | 3       | 4531.1 | 3       | 16.7  |
|        | 20     | 3       | 775.5  | 3       | 13.8  | 3       | 2358.8 | 3       | 15.0  |
|        | 25     | 3       | 1494.0 | 3       | 16.8  | 3       | 1472.0 | 3       | 16.8  |
|        | 30     | 3       | 2158.0 | 3       | 16.8  | 3       | 2088.0 | 3       | 15.0  |

**Table 4.** Running time (seconds) of the Lanczos method using the CG/PCG method in the case of the 5-th max. singular value of almost symmetric matrix $T$ with $a = c = 1$ and $b = 0.01$ in (9) for the shift (8).

|        | Method | Algorithms 1 and 3 (by Eigendecompn.) | | Algorithms 1 and 4 (by Schur Decompn.) | |
|--------|--------|-------------------|--------------------|-------------------|--------------------|
|        |        | Lanczos with CG   | Lanczos with PCG   | Lanczos with CG   | Lanczos with PCG   |
|        | 5      | 0.114             | 0.071              | 0.095             | 0.061              |
|        | 10     | 0.578             | 0.095              | 0.566             | 0.086              |
| $n$    | 15     | 1.402             | 0.137              | 1.448             | 0.192              |
|        | 20     | 6.639             | 0.437              | 6.446             | 0.335              |
|        | 25     | 13.632            | 0.558              | 12.686            | 0.432              |
|        | 30     | 35.121            | 1.145              | 33.203            | 0.836              |

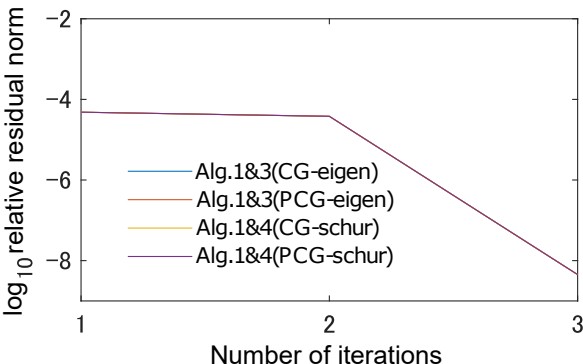

**Figure 1.** Convergence histories with relative residual norm of the Lanczos method for the 5-th max. singular value of the almost symmetric matrix $T$ whose size is $n = 15$.

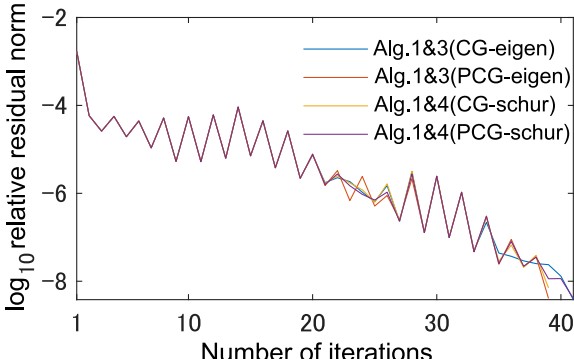

**Figure 2.** Convergence histories with relative residual norm of the Lanczos method for the median singular value of the almost symmetric matrix $T$ whose size is $n = 15$.

Next, we show the second results in the case of slightly symmetric matrix with $a = c = 1$ and $b = 0.1$ in Equation (9) for the shift (8). From Table 5, both PCG methods did not converge for computing the 5-th maximum singular values of slightly symmetric matrix $T$. It seems that the linear system for $T^{\mathsf{T}}T - \tilde{\sigma}^2 I_{\ell mn}$ is ill-conditioned since $10^{-2}$ in the shift (8) is much less than the 5-th maximum singular values of the matrix. In Appendix A, we show the results using relative shift without the effect of the magnitude of the singular values. Table 6 shows Algorithms 1 and 4, that is, the algorithms based on Schur decomposition, was more robust than Algorithms 1 and 3, that is, the algorithms based on the eigendecomposition. From Table 7, both PCG methods converged regardless of $n$, and the numbers of iterations of both PCG methods were less than the number of iterations of both CG method. Namely, it seems that the preconditioning matrix $M$ can be effective in the case of a slightly symmetric matrix $T$ when computing the 5-th minimum and median singular values of $T$.

**Table 5.** Number of iterations of the Lanczos method and the average of numbers of iterations of the CG/PCG method in the case of the 5-th max. singular value of slightly symmetric matrix $T$ with $a = c = 1$ and $b = 0.1$ in (9) for the shift (8).

| Method | | Algorithms 1 and 3 (by Eigendecompn.) | | | | Algorithms 1 and 4 (by Schur Decompn.) | | | |
|---|---|---|---|---|---|---|---|---|---|
| | | Lanczos | CG | Lanczos | PCG | Lanczos | CG | Lanczos | PCG |
| | 5 | 4 | 50.0 | (Not converged.) | | 4 | 37.8 | (Not converged.) | |
| | 10 | 4 | 100.0 | (Not converged.) | | 4 | 87.3 | (Not converged.) | |
| | 15 | 3 | 152.0 | (Not converged.) | | 3 | 131.0 | (Not converged.) | |
| $n$ | 20 | 4 | 205.5 | (Not converged.) | | 4 | 172.0 | (Not converged.) | |
| | 25 | 3 | 262.0 | (Not converged.) | | 3 | 230.0 | (Not converged.) | |
| | 30 | 3 | 306.7 | (Not converged.) | | 3 | 259.0 | (Not converged.) | |

**Table 6.** Number of iterations of the Lanczos method and the average of numbers of iterations of the CG/PCG method in the case of the median of singular value of slightly symmetric matrix $T$ with $a = c = 1$ and $b = 0.1$ in (9) for the shift (8).

| Method | | Algorithms 1 and 3 (by Eigendecompn.) | | | | Algorithms 1 and 4 (by Schur Decompn.) | | | |
| | | Lanczos | CG | Lanczos | PCG | Lanczos | CG | Lanczos | PCG |
|---|---|---|---|---|---|---|---|---|---|
| | 5 | 13 | 231.8 | (Not converged.) | | 13 | 193.1 | 13 | 73.0 |
| | 10 | 6 | 1582.7 | 6 | 55.0 | 6 | 1272.7 | 6 | 29.0 |
| *n* | 15 | 30 | 12,174.6 | (Not converged.) | | 31 | 11061.9 | 31 | 97.0 |
| | 20 | 4 | 13,777.8 | (Not converged.) | | 4 | 8799.3 | (Not converged.) | |
| | 25 | (Not converged.) | | (Not converged.) | | (Not converged.) | | 83 | 116.0 |
| | 30 | (Not converged.) | | (Not converged.) | | (Not converged.) | | 27 | 45.0 |

**Table 7.** Number of iterations of the Lanczos method and the average of numbers of iterations of the CG/PCG method in the case of the 5-th min. singular value of slightly symmetric matrix $T$ with $a = c = 1$ and $b = 0.1$ in (9) for the shift (8).

| Method | | Algorithms 1 and 3 (by Eigendecompn.) | | | | Algorithms 1 and 4 (by Schur Decompn.) | | | |
| | | Lanczos | CG | Lanczos | PCG | Lanczos | CG | Lanczos | PCG |
|---|---|---|---|---|---|---|---|---|---|
| | 5 | 3 | 195.5 | 3 | 59.0 | 3 | 163.9 | 3 | 68.4 |
| | 10 | 3 | 949.8 | 3 | 58.0 | 3 | 771.2 | 3 | 44.0 |
| *n* | 15 | 3 | 616.0 | 3 | 63.0 | 3 | 597.3 | 3 | 94.5 |
| | 20 | 3 | 859.8 | 3 | 63.0 | 3 | 2999.8 | 3 | 57.0 |
| | 25 | 3 | 1695.7 | 3 | 63.0 | 3 | 1559.3 | 3 | 114.4 |
| | 30 | 3 | 2388.0 | 3 | 63.0 | 3 | 2262.0 | 3 | 46.1 |

Finally, we show the third results in the case of marginally symmetric matrix with $a = c = 1$ and $b = 0.2$ in Equation (9) for the shift (8). Both PCG methods did not converge for computing the 5-th maximum singular values of $T$ as shown in Table 8, similarly to Table 5. Moreover, computing the median singular values of $T$ sometimes did not converge from Table 9. In Table 10, all methods converged for the 5-th minimum singular value of $T$. The numbers of iterations by the PCG method with the proposed preconditioning matrix were less than the number of iterations by the CG method in most cases. It seems that the preconditioning matrix $M$ can be effective in the case of the marginally symmetric matrix $T$ when computing the 5-th minimum singular values of $T$.

**Table 8.** Number of iterations of the Lanczos method and the average of numbers of iterations of the CG/PCG method in the case of the 5-th max. singular value of marginally symmetric matrix $T$ with $a = c = 1$ and $b = 0.2$ in (9) for the shift (8).

| Method | | Algorithms 1 and 3 (by Eigendecompn.) | | | | Algorithms 1 and 4 (by Schur Decompn.) | | | |
| | | Lanczos | CG | Lanczos | PCG | Lanczos | CG | Lanczos | PCG |
|---|---|---|---|---|---|---|---|---|---|
| | 5 | 4 | 52.8 | (Not converged.) | | 4 | 39.8 | (Not converged.) | |
| | 10 | 4 | 107.0 | (Not converged.) | | 4 | 93.0 | (Not converged.) | |
| *n* | 15 | 3 | 162.0 | (Not converged.) | | 3 | 135.7 | (Not converged.) | |
| | 20 | 4 | 217.8 | (Not converged.) | | 4 | 206.3 | (Not converged.) | |
| | 25 | 3 | 271.3 | (Not converged.) | | 3 | 248.0 | (Not converged.) | |
| | 30 | 3 | 334.0 | (Not converged.) | | 3 | 260.0 | (Not converged.) | |

**Table 9.** Number of iterations of the Lanczos method and the average of numbers of iterations of the CG/PCG method in the case of the median of singular value of marginally symmetric matrix *T* with $a = c = 1$ and $b = 0.2$ in (9) for the shift (8).

| Method | Algorithms 1 and 3 (by Eigendecompn.) | | | | Algorithms 1 and 4 (by Schur Decompn.) | | | |
| --- | --- | --- | --- | --- | --- | --- | --- | --- |
| | Lanczos | CG | Lanczos | PCG | Lanczos | CG | Lanczos | PCG |
| 5 | 11 | 481.4 | (Not converged.) | | 10 | 355.6 | (Not converged.) | |
| 10 | 6 | 2123.8 | (Not converged.) | | 10 | 1550.8 | 10 | 4262.4 |
| *n* = 15 | (Not converged.) | | (Not converged.) | | (Not converged.) | | (Not converged.) | |
| 20 | 15 | 13,268.7 | 89 | 6358.9 | 7 | 10019.9 | 108 | 160.0 |
| 25 | (Not converged.) | | (Not converged.) | | (Not converged.) | | (Not converged.) | |
| 30 | (Not converged.) | | 90 | 1150.0 | (Not converged.) | | 28 | 11,764.0 |

**Table 10.** Number of iterations of the Lanczos method and the average of numbers of iterations of the CG/PCG method in the case of the 5-th min. singular value of marginally symmetric matrix *T* with $a = c = 1$ and $b = 0.2$ in (9) for the shift (8).

| Method | Algorithms 1 and 3 (by Eigendecompn.) | | | | Algorithms 1 and 4 (by Schur Decompn.) | | | |
| --- | --- | --- | --- | --- | --- | --- | --- | --- |
| | Lanczos | CG | Lanczos | PCG | Lanczos | CG | Lanczos | PCG |
| 5 | 5 | 313.9 | 5 | 295.8 | 5 | 213.9 | 5 | 1077.8 |
| 10 | 3 | 1247.8 | 3 | 187.0 | 3 | 1150.6 | 3 | 3048.8 |
| *n* = 15 | 3 | 650.0 | 3 | 201.0 | 3 | 606.7 | 3 | 177.0 |
| 20 | 3 | 929.3 | 3 | 6151.3 | 3 | 847.8 | 3 | 162.8 |
| 25 | 3 | 1800.3 | 3 | 205.0 | 3 | 1683.7 | 3 | 249.0 |
| 30 | 3 | 2585.7 | 3 | 1118.6 | 3 | 2371.3 | 3 | 181.0 |

## 6. Conclusions

We considered computing an arbitrary singular value of a tensor sum. The shift-and-invert Lanczos method and the PCG method reconstructed over tensor space. We proposed the preconditioning matrices which are the following two diagonal matrices: (1) whose diagonals of the eigenvalues by the eigendecomposition, and (2) whose diagonals of the upper diagonal matrix by the Schur decomposition. This preconditioning matrix can be effective if the tensor sum is almost symmetric.

From numerical results, we confirmed that the proposed method reduces memory requirements without any conditions. The numbers of iterations of the PCG method by the proposed preconditioning matrix were reduced in most cases of the almost and slightly symmetric matrix. Moreover, the numbers of iterations of the PCG method by the proposed preconditioning matrix were also reduced in certain cases of the marginally symmetric matrix.

For future work, we will consider a robust preconditioning matrix for slightly or marginally symmetric tensor sum, a suitable preconditioning matrix for non-symmetric tensor sum, parallel implementations of the proposed algorithms, and finding real-life applications.

**Author Contributions:** Conceptualization, A.O. and T.S.; investigation, A.O. and T.S.; software, A.O.; validation, T.S.; writing—original draft, A.O.; writing—review and editing, A.O. and T.S. All authors have read and agreed to the published version of the manuscript.

**Funding:** This work was supported by JSPS KAKENHI Grant Numbers: JP21K17754, JP20H00581, JP20K20397, JP17H02829.

**Institutional Review Board Statement:** Not applicable.

**Informed Consent Statement:** Not applicable.

**Data Availability Statement:** Not applicable.

**Conflicts of Interest:** The authors declare no conflict of interest.

## Abbreviations

The following abbreviations are used in this manuscript:

PCG    Preconditioned Conjugate Gradient
CG     Conjugate Gradient
PDE    Partial Differential Equation

## Appendix A

This appendix gives the numerical results in the case of the 5-th maximum and the median singular values of slightly and marginally symmetric matrices by the relative shift

$$\tilde{\sigma} = \sigma - 10^{-2}\sigma, \tag{A1}$$

where $\sigma$'s are the singular values of $T$ computed by the svd function in MATLAB. The condition of the numerical experiments except for the setting of the shift is the same as the experiments in Section 5.

Firstly, we show the results in the case of slightly symmetric matrix with $a = c = 1$ and $b = 0.1$ in Equation (9) for the shift (A1) in Tables A1 and A2. Computing the 5-th and the median singular values of the slightly symmetric matrix using the shift (A1), the number of iterations of both PCG methods is much less than the number of iterations of both CG methods.

Secondly, Tables A3 and A4 are the results in the case of marginally symmetric matrix with $a = c = 1$ and $b = 0.2$ in Equation (9) for the shift (A1). From Tables A3 and A4, both PCG methods converged faster than both CG method using the relative shift. Moreover, the PCG method by Algorithm 4 is more robust than the PCG method by Algorithm 3.

Consequently, when we compute the 5-th maximum and the median singular values of the slightly symmetric matrix, the numerical experiments of Section 5 and Appendix A imply that the proposed preconditioning matrix can work in the case of a suitable shift.

**Table A1.** Number of iterations of the Lanczos method and the average of numbers of iterations of the CG/PCG method in the case of the 5-th max. singular value of slightly symmetric matrix $T$ with $a = c = 1$ and $b = 0.1$ in (9) for the shift (A1).

| Method | | Algorithms 1 and 3 (by Eigendecompn.) | | | | Algorithms 1 and 4 (by Schur Decompn.) | | | |
|---|---|---|---|---|---|---|---|---|---|
| | | Lanczos | CG | Lanczos | PCG | Lanczos | CG | Lanczos | PCG |
| | 5 | 5 | 41.0 | 5 | 17.0 | 5 | 35.8 | 5 | 15.0 |
| | 10 | 7 | 84.0 | 7 | 20.0 | 7 | 77.0 | 7 | 13.0 |
| *n* | 15 | 4 | 162.0 | 4 | 22.0 | 4 | 154.0 | 4 | 17.0 |
| | 20 | 7 | 223.7 | 7 | 23.0 | 7 | 197.3 | 7 | 12.0 |
| | 25 | 5 | 383.6 | 5 | 24.0 | 5 | 307.4 | 5 | 15.0 |
| | 30 | 6 | 522.7 | 6 | 24.0 | 6 | 426.5 | 6 | 13.0 |

**Table A2.** Number of iterations of the Lanczos method and the average of numbers of iterations of the CG/PCG method in the case of the median of singular value of slightly symmetric matrix $T$ with $a = c = 1$ and $b = 0.1$ in (9) for the shift (A1).

| Method | | Algorithms 1 and 3 (by Eigendecompn.) | | | | Algorithms 1 and 4 (by Schur Decompn.) | | | |
|---|---|---|---|---|---|---|---|---|---|
| | | Lanczos | CG | Lanczos | PCG | Lanczos | CG | Lanczos | PCG |
| | 5 | 23 | 139.3 | 23 | 38.0 | 23 | 105.9 | 23 | 14.0 |
| | 10 | 10 | 1081.0 | 10 | 21.0 | 10 | 1074.4 | 10 | 22.0 |
| *n* | 15 | 21 | 5201.6 | 21 | 21.0 | 21 | 3470.4 | 21 | 14.0 |
| | 20 | 17 | 7333.1 | (Not converged.) | | 17 | 6242.4 | 17 | 16.0 |
| | 25 | 11 | 16,034.1 | 11 | 32.0 | 11 | 14,360.6 | 11 | 14.0 |
| | 30 | (Not converged.) | | 12 | 23.0 | (Not converged.) | | 12 | 17.0 |

**Table A3.** Number of iterations of the Lanczos method and the average of numbers of iterations of the CG/PCG method in the case of the 5-th max. singular value of marginally symmetric matrix $T$ with $a = c = 1$ and $b = 0.2$ in (9) for the shift (A1).

| Method | | Algorithms 1 and 3 (by Eigendecompn.) | | | | Algorithms 1 and 4 (by Schur Decompn.) | | | |
| --- | --- | --- | --- | --- | --- | --- | --- | --- | --- |
| | | Lanczos | CG | Lanczos | PCG | Lanczos | CG | Lanczos | PCG |
| | 5 | 5 | 43.0 | 5 | 25.0 | 5 | 38.4 | 5 | 17.0 |
| | 10 | 7 | 90.0 | 7 | 29.0 | 7 | 79.0 | 7 | 13.0 |
| $n$ | 15 | 4 | 174.5 | 4 | 32.0 | 4 | 152.5 | 4 | 62.0 |
| | 20 | 7 | 253.0 | 7 | 33.0 | 7 | 198.1 | 7 | 15.0 |
| | 25 | 5 | 403.0 | 5 | 35.0 | 5 | 319.0 | 5 | 19.0 |
| | 30 | 6 | 626.0 | 6 | 36.0 | 6 | 441.2 | 6 | 16.0 |

**Table A4.** Number of iterations of the Lanczos method and the average of numbers of iterations of the CG/PCG method in the case of the median of singular value of marginally symmetric matrix $T$ with $a = c = 1$ and $b = 0.2$ in (9) for the shift (A1).

| Method | | Algorithms 1 and 3 (by Eigendecompn.) | | | | Algorithms 1 and 4 (by Schur Decompn.) | | | |
| --- | --- | --- | --- | --- | --- | --- | --- | --- | --- |
| | | Lanczos | CG | Lanczos | PCG | Lanczos | CG | Lanczos | PCG |
| | 5 | 24 | 138.3 | (Not converged.) | | 25 | 115.6 | 23 | 17.0 |
| | 10 | 10 | 1479.0 | (Not converged.) | | 10 | 1119.2 | 10 | 18.0 |
| $n$ | 15 | 21 | 4506.6 | 21 | 34.0 | 21 | 3787.0 | 21 | 16.0 |
| | 20 | 17 | 8603.3 | (Not converged.) | | 17 | 6604.5 | 17 | 21.0 |
| | 25 | 11 | 18,267.0 | (Not converged.) | | 11 | 14,991.6 | 11 | 30.0 |
| | 30 | (Not converged.) | | 13 | 35.0 | (Not converged.) | | 13 | 31.0 |

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
