# Peer review of "Numerical Algorithms for Computing an Arbitrary Singular Value of a Tensor Sum"

_axioms, doi:10.3390/axioms10030211_

Round 1

Reviewer 1 Report

The authors tackle the computation of an arbitrary singular values of a tensor sum (T), supported in previous work by the same authors in the calculation of extreme singular values. Two approaches are offered: (i) shift-and-invert Lanczos method for the eigenvalue problem T^TT and the PCG method for the solution of T^TT-\tilde{sigma}^2I and (ii preconditioned CG over the tensor space with Schur decomposition of T.

Numerical experiments are performed on test matrices obtained from a seven-point central difference discretization of a PDE.

Major comments:

  • What do the authors intend to highlight with “Since the direct methods cannot be applied due to the nonzero structure of the coefficient matrix”? Why does the structure of the coefficient matrix does not allow for the use of a direct method approach?
  • For the numerical results, it is not clear for the reader the choice of the shifts. A justification for this choice is expected. Was a different shift used for each one of the singular values to be pursued (even if may be obvious, let it clear)? Why was the parcel 10^-2 used for all cases? Please clarify this.
  • The sentence “… or including NaN (…) in variables of these algorithms” is not noticeable. On the other hand, in none of the results reported does this designation of NaN appear!
  • The saw effect visible in the graphs of Figure 2 deserves an interpretation. The result is not expected, so a possible justification must be presented, or a more detailed analysis becomes essential.
  • Regarding Tables 1 to 3, the authors should comment on the different number of iterations required by the method for the three sought singular values: namely, why is it larger for the median singular value? Is it problem dependent or depends on the worse choice for the shift in this case?
  • Results in Tables 4 to 6 are disturbing. The preconditioner did not allow to reach convergence for the 5th largest and median singular value, contrary to the 5th minimum? Similarly in the case of marginally symmetric matrix, Tables 8, 9 and 10. Results for the 5th minimum singular value are expected, and a sentence commenting on this is also to be expected. The other results cannot be presented without a proper justification. Wouldn't it be the case that the shifts were not properly chosen, or the preconditioning matrix wrongly implemented? Why such a simple approach to M, which may not give rise to a huge improvement in the condition number to the operator matrix, avoids the method to converge?
  • Why the authors did not report on the elapsed times of the methods proposed (as in previous publications on the same topic)? the reader would have a better understanding of the methods, in addition to allowing the two approaches proposed in terms of execution speed to be compared.
  • Additional examples are to be expected, and not only rely on one example to get your message across.

Minor comments:

Line 20: remove “too large”

Line 22: Consider “Previous” instead of “The previous”

Line 22: Consider “methods” instead of “the methods”

Line 23: “are” instead of “were”

Line 25: I would suggest removing the sentence “Here, …of a matrix.”

Line 26: “For insights on” instead of “For the development of”

Line 40: “requires the solution” instead of “needs a solver”

Line 43: consider “Even though it is difficult …”

Lines 95-97: consider “So, even if $T$ is not exactly

symmetric, if $T$ is almost symmetric then we can expect the preconditioning matrix M to be effective.”

Line 139: consider “… criteria used in Algorithm 1…”

Line 139: remove “we used”

Line 140: consider “… is the eigenvector …”

Line 158: consider “… were almost the same”

Author Response

I attached the reply to the referee's report of reviewer #1.

Reviewer 2 Report

The article "Numerical algorithms for calculating an arbitrary singular value of the tensor sum" discusses the issues of calculating an arbitrary singular value of the tensor sum using the Lanczos method with shift and inversion, as well as the preliminary conjugate gradient method. Algorithms for calculating the tensor sum are proposed, which have been investigated on specific examples and have proven their effectiveness.

Undoubtedly, the work is relevant, has a wide practical application. However, there are some guidelines that can improve this article:

  1. All abbreviations must be placed in a separate table at the end of the article.
  2. Due to the fact that the authors work with matrices of large dimension, it is necessary to show the running time of the proposed algorithms. The authors showed that the number of iterations decreases with the help of computational procedures, but the algorithm can work for a rather long time.
  3. The second question follows from the first. Are parallel implementations of the proposed algorithms possible? If yes, then this fact must be indicated in the article.

In general, the article can be recommended for publication after minor adjustments.

Author Response

I attached the reply to the referee's report of reviewer #2.

Reviewer 3 Report

This is a study on tensor sums.

Strenghs: (1)The Lanczos-like method is new and shown to be efficient for computing  a singular value of a tensor program.

(2)The algorithm is well developed and explained. 

Weaknesses:(1)State benefits over using other methods already in the literature.

(2)Cite or delete non cited references.

(3) Suggest real life applications for this method.  

Author Response

I attached the reply to the referee's report of reviewer #3.

Round 2

Reviewer 1 Report

The authors answered and detailed the most important recommendations offered. This revised version has been greatly improved over the previous.

Reviewer 3 Report

Changes made.